# Parents’ Intentions, Concerns and Information Needs about COVID-19 Vaccination in New Jersey: A Qualitative Analysis

**DOI:** 10.3390/vaccines11061096

**Published:** 2023-06-13

**Authors:** Racquel E. Kohler, Rachel B. Wagner, Katherine Careaga, Jacqueline Vega, Rula Btoush, Kathryn Greene, Leslie Kantor

**Affiliations:** 1Center for Cancer Heath Equity, Rutgers Cancer Institute of New Jersey, New Brunswick, NJ 08901, USAklgreene@rutgers.edu (K.G.); 2School of Public Health, Rutgers University, Piscataway, NJ 08854, USA; lk481@sph.rutgers.edu; 3School of Nursing, Rutgers University, New Brunswick, NJ 08901, USA; 4School of Communication & Information, Rutgers University, New Brunswick, NJ 08901, USA

**Keywords:** COVID-19, vaccine hesitancy, vaccination behavior, health communication, adolescent health, parents

## Abstract

Background: In 2019, the World Health Organization identified vaccine hesitancy as a top ten global health threat, which has been exacerbated by the COVID-19 pandemic. Despite local and nationwide public health efforts, adolescent COVID-19 vaccination uptake in the US remains low. This study explored parents’ perceptions of the COVID-19 vaccine and factors influencing hesitancy to inform future outreach and education campaigns. Methods: We conducted two rounds of individual interviews via Zoom in May–September 2021 and January–February 2022, with parents of adolescents from the Greater Newark Area of New Jersey, a densely populated area with historically marginalized groups that had low COVID-19 vaccination uptake. Data collection and analysis was guided by the Increasing Vaccination Model and WHO Vaccine Hesitancy Matrix. Interview transcripts were double-coded and thematically analyzed in NVivo. Results: We interviewed 22 parents (17 in English, 5 in Spanish). Nearly half (45%) were Black and 41% were Hispanic. Over half (54%) were born outside of the US. Most of the parents described that their adolescents had received at least one dose of a COVID-19 vaccine. All but one parent had received the COVID-19 vaccine. Despite strong vaccination acceptance for themselves, parents remained hesitant about vaccinating their adolescents. They were mostly concerned about the safety and potential side effects due to the novelty of the vaccine. Parents sought information about the vaccines online, through healthcare providers and authorities, and at community spaces. Interpersonal communication exposed parents to misinformation, though some personal connections to severe COVID-19 illness motivated vaccination. Historical mistreatment by the healthcare system and politicization of the vaccine contributed to parents’ mixed feelings about the trustworthiness of those involved with developing, promoting, and distributing COVID-19 vaccines. Conclusions: We identified multilevel influences on COVID-19 vaccine-specific hesitancy among a racially/ethnically diverse sample of parents with adolescents that can inform future vaccination interventions. To increase vaccine confidence, future COVID booster campaigns and other vaccination efforts should disseminate information through trusted healthcare providers in clinical and also utilize community settings by addressing specific safety concerns and promoting vaccine effectiveness.

## 1. Introduction

Vaccines are among the world’s greatest public health advancements and are crucial to the prevention and control of infectious disease outbreaks [1,2]. Childhood vaccination prevented four million deaths in 2021 and another 51 million deaths are projected to be averted through vaccination between 2021 and 2030 [3,4]. Despite long-standing, strong evidence promoting efficacious and safe vaccines, uptake is increasingly specific to a particular vaccine or preventable disease [5,6]. The reasons that individuals delay or reject vaccination are complex [7]. Vaccine hesitancy, defined as a reluctance or uncertain attitude toward vaccination, may apply to a specific vaccine or to vaccines in general [8,9,10]. Vaccine hesitancy can influence behaviors such as accepting, delaying, or refusing vaccination [9,11]. In 2019, the World Health Organization (WHO) identified vaccine hesitancy as one of the top ten global health threats [12], which has been exacerbated by COVID-19 vaccination. Vaccine hesitancy is also context-specific and therefore varies by time and geographic location [13]. Understanding local determinants of vaccine-specific hesitancy is particularly important in developing communication strategies to promote initial vaccination and boosters for emerging diseases, such as COVID-19 [5,14]. Survey data from before and after COVID-19 vaccine trials for children found that parents were very hesitant about COVID-19 vaccination [15,16]. Once it was recommended for adolescents, vaccination uptake and completion varied by state and age group (higher coverage among older teenagers) [17]. 

In addition to national public health efforts, state and local Departments of Health (DOH) launched COVID-19 public awareness campaigns, providing training, toolkits, and digital resources on COVID-19 testing, contact tracing, and vaccination for both children and adults [18,19]. In New Jersey (NJ), the DOH implemented a COVID-19 Information Hub website, with regular updates on COVID-19 cases and vaccination statistics, as well as testing and vaccination location information and appointment requests. Other statewide COVID-19 initiatives created YouTube videos debunking COVID-19 misinformation and SMS text and social media campaigns in both English and Spanish [20,21]. In October 2022, COVID-19 vaccine coverage for two doses in NJ (77.4%) exceeded the US (67.8%) [22] and the WHO global goal of 70% uptake [23]. However, initial booster uptake in October 2022 remained low at only 50% statewide [22]. The bivalent booster was approved in September 2022 and, as of March 2023, bivalent booster uptake in NJ was 16.8% [22], in line with the national average of 16.7%. In addition to increasing access to vaccination [24], public health messaging tailored to local factors influencing vaccine hesitancy is needed to increase booster uptake for the bivalent booster, as well as vaccines in future years [25,26,27]. 

Although the Advisory Committee on Immunization Practices made interim recommendations for vaccination under emergency use authorization (EUA) for those aged 6 to 18 years [28], the COVID-19 vaccine is not mandated for students to attend school in NJ [29]. In August 2022, Moderna and Pfizer/BioNTech boosters were authorized for children 12 years and older, and in January 2023, COVID-19 was added to the recommended routine vaccination schedule for all children 6 months and older [30]. Although severe illness and hospitalization are less common among children compared to adults, over 13 million children have contracted COVID-19 and can spread the virus [31]. Long-term sequelae of multiple infections are only beginning to emerge. As prevention strategies and policies for school-age children vary greatly, understanding parents’ concerns and motivations about vaccination is essential to inform strategies for increasing uptake [8]. Therefore, we conducted a qualitative study to assess parents’ perceptions of COVID-19 vaccination for adolescents to inform outreach and vaccination communication strategies. We explored vaccination concerns and information needs specific to the parents living in the Greater Newark Area of NJ, a densely populated area with a high percentage of historically marginalized groups, including Blacks, Hispanics, and other individuals of color, that had low COVID vaccination uptake [22,32,33].

## 2. Methods

### 2.1. Study Setting and Design

As part of a larger mixed-methods study assessing vaccine hesitancy in the Greater Newark Area, this qualitative analysis focused on perceptions of COVID-19 vaccination among parents of adolescents in Essex and Hudson Counties. We recruited parents within this area because of lower rates of adolescent vaccination [34]. We recruited parents through existing partnerships with local community-based organizations and agencies. We recruited at public libraries, food pantries, elementary and middle schools, health fairs, and family events in both counties. English- and Spanish-speaking parents of adolescents aged 9 to 13 years of age were eligible if they were a “healthcare decision maker” for the adolescent and resided in the study communities. While parents were asked to focus on their adolescents aged 9 to 13 for the interview, some had multiple children of different ages and included them in their narratives.

Individual interviews were conducted during COVID-19 vaccine development, including active clinical trials, emergency use authorizations (EUAs), and FDA approvals. Specifically, the Pfizer vaccine was under EUA for 12 years and older for the entire study period (May 2021 to February 2022); the EUA was expanded to 5–11 years mid-way in October 2021. Therefore, our study period provided an opportunity to capture participants’ reactions to these events in real time and how they influenced vaccine attitudes and behaviors at different points during the pandemic. Figure 1 depicts the data collection timeline with participant vaccination behaviors and vaccine approval dates for Pfizer and Moderna. We had two recruitment periods: 12 participants were interviewed between May and September 2021, and 10 participants were interviewed between January and February 2022. We aimed to recruit parents who were both from different racial/ethnic backgrounds and with adolescents vaccinated and unvaccinated. We conducted 22 interviews in both English (n = 17) and Spanish (n = 5).

### 2.2. Conceptual Model

Data collection and analysis for the overall study was guided by the Increasing Vaccination Model [35] and the WHO Vaccine Working Group’s Vaccine Hesitancy Determinants Matrix [36,37]. For this study, we considered COVID-19 vaccination behaviors (i.e., receipt or refusal) to be influenced by motivation, or hesitancy, to vaccinate. Therefore, we characterize COVID-19 vaccine hesitancy as a reluctant attitude toward vaccination, which is predicted by the perceived risk of COVID-19 and confidence in vaccine effectiveness and safety. Because of the evolution of EUAs, eligible age groups, and vaccination delivery, we also considered parents’ intentions to vaccinate and how they could be influenced by convenience, including logistical issues or structural barriers to vaccination. 

### 2.3. Data Collection and Analysis 

Eligible participants provided written informed consent and completed a brief online intake survey that collected sociodemographic data, insurance status, and immunization history. Parents were invited to a HIPAA-compliant Zoom room where female study staff trained in anthropology and health psychology (KC and RW) conducted individual interviews in English or Spanish based on each participant’s preference, which facilitated a diverse variety of parent perspectives. Data collection ended when data saturation was reached, when no new emerging themes occurred during interviews [38]. 

The semi-structured interview guide included a COVID-19 section composed of questions related to COVID-19 vaccination attitudes, beliefs, and behaviors or intentions of participants for themselves and their families. Examples of the questions asked are as follows: (1) Where/who do you go to for information about COVID-19 vaccination? (2) Has your adolescent received the COVID-19 vaccine? Why or why not? Do you plan to vaccinate them? (3) How have your views of vaccines changed since the beginning of the COVID-19 pandemic? (4) What do you think about the government’s role in COVID vaccination? The online interviews were 45 min on average. 

All recordings were professionally transcribed, de-identified, and verified for accuracy by research staff. The multidisciplinary study team reviewed transcripts and met regularly to discuss analysis. Two bilingual staff (KC and JV) fluent in Spanish analyzed the Spanish transcripts and then translated key text for combined analysis in English. All transcripts were double-coded by team members (RW, KC, JV) in NVivo 2020 and thematically analyzed using an iterative immersion–crystallization approach [39,40]. The analytic team (REK, RW, KC, JV) conducted multiple rounds of reading transcripts, writing summaries, identifying themes, coding and organizing text, reflecting on bias and experience, synthesizing patterns, and corroborating interpretations. The analysis was guided by the conceptual frameworks to identify how thoughts and feelings influenced intentions and behaviors [35] and how complex factors across multiple socioecological levels impacted vaccine development, promotion, and delivery [41]. Themes emerging from the data were also identified and included as new analytic codes/concepts. The full study team with additional experts in nursing, communication, and public health (RB, KG, LK) also discussed themes and patterns, making comparisons and checking assumptions. 

## 3. Results

### 3.1. Sample Characteristics and Key Themes

We enrolled 22 female parents of adolescents (Table 1). Nearly half of the parents (45%) were Black, and 41% were Hispanic. Fifty-five percent of the sample were born outside of the US, including countries in Latin America, the Caribbean, South America, Asia, and Africa. Similar to the state’s overall high educational attainment [32], many participants had attended college. However, five (23%) had a high school diploma or did not complete high school; all of these participants were born outside of the US and completed the interview in Spanish. Many parents (86%) had multiple children under 18 living in the household and sometimes included these other children in their narratives. The mean age of the targeted adolescents was 11 years. Most adolescents had received at least one dose of a COVID-19 vaccine. Half of the parents reported that their adolescent was covered under Medicaid or CHIP. 

Below, we describe COVID-19 vaccination behaviors and intentions and how they were shaped by individual, interpersonal, and contextual factors. Despite concerns about vaccine safety and side effects, parents found the high-efficacy data about COVID-19 vaccines compelling. Some parents were exposed to misinformation through interpersonal attitudes, but they relied on trusted information sources to make vaccination decisions. Historical, governmental, and political contexts influenced parents’ perceptions of vaccination and mandates, which were also influenced by poor COVID-19 communication. We present quotes that illustrate constructs from our conceptual model in Table 2.

### 3.2. High Acceptance and Strong Intentions to Vaccinate 

Our sample was highly accepting of COVID-19 vaccines and had vaccinated themselves and their children. We observed increasing acceptance of COVID-19 vaccination for adolescents in interviews taking place over time (Figure 1). Most parents (n = 18) reported being “fully vaccinated”, though a few (n = 3) had received at least one dose. One additional parent from the first round of interviews did not accept the vaccine for herself or child, noting that she was still “go[ing] back and forth” and wanted to “do more research about it.” Parents typically talked about vaccination status in terms of their household, stating which family members were vaccinated or awaiting vaccination based on age eligibility. No parents reported any spousal or co-parent discrepancies or disagreements on vaccinations within their household. Many of the participants had both older teenagers and younger children and shared that all of their age-eligible children were also vaccinated against COVID-19. In addition to over half of the parents reporting that their child was vaccinated with at least one dose of the vaccine, many of those with unvaccinated children intended to accept the vaccine “right after it was approved” for their child’s age group. One parent was particularly enthusiastic about vaccination: 

“I love them. I had my third shot. My wife had her third shot. My son has been vaccinated. My daughter’s been vaccinated.”P12, parent of a 13-year-old male in September 2021

### 3.3. Safety and Side Effect Concerns Due to the Vaccine’s Novelty Dominated Hesitancy Discussions 

Despite being generally positive toward vaccination for their adolescents, parents reported that the novelty and rapid development of the COVID-19 vaccine made them perceive it as unsafe. Most parents felt that the vaccine development and approval processes were rushed, which intensified their concerns about side effects associated with a new vaccine. When comparing COVID-19 to other recommended vaccines for their children, parents felt that the other vaccines were safer because they had “enough research [from] people that have taken” the vaccines and they had been “studied long enough in enough kids.” The most common concerns were long-term infertility, hindering pre-pubescent development, and short-term side effects or reactions: 

“Since it’s new and my daughter is going through this development stage, I was a little, a little afraid. I guess a lot of parents are right now with all these vaccines for COVID. And my son has allergies, so that’s why it was a little scary just, thinking he might have a reaction from it.”P16, parent of an 11-year-old female in January 2022

Even though some parents raised concerns about how the vaccine might interact with their adolescents’ comorbidities (e.g., autism, Down’s Syndrome, or heart conditions), they ultimately vaccinated their adolescents. As the interviews progressed, all parents’ narratives shifted from questioning the safety of the vaccine to potentially jeopardizing their adolescents’ health by delaying or refusing it.

### 3.4. Interpersonal Misinformation and Motivations 

In comparison to other vaccines, parents had more interpersonal conversations about the COVID-19 vaccine within their social networks, mainly with other parents, friends, and family members. Participants also described how they perceived vaccination patterns within their networks. For example, they noticed that parents of pre-pubescent girls preferred to wait until after puberty to vaccinate, compared to those with daughters who had already started puberty, who were generally more comfortable vaccinating their adolescents. Other parents mentioned that their friends had warned them against vaccination because it may cause side effects such as infertility. Some reported distant family members that relocated from New Jersey to Florida in order to avoid any potential school vaccine requirements for their children, including for COVID-19. 

However, participants tended to rely on trustworthy sources for vaccine information over their families’ and friends’ opinions. They also recalled close encounters with severe COVID-19 infections or deaths that influenced their risk appraisal. For example, one parent shared how she was influenced by different peers and her perceptions shifted over time:

“But I’m gonna be honest, at first, I was against the vaccine. I didn’t want to take it. I refused it […] listening to bogus people […] I’m talking about my friends. I’m talking about some coworkers. ‘I’m not taking that vaccine. They’re trying to kill me.’ That’s what I was thinking at first, but then I started listening to the doctors, thinking about my own health, hearing stories of people that I knew that had COVID that were in the hospital that was on those machines, and I know that’s real.”P05, parent of a 10-year-old male in July 2021

Two non-US-born, Hispanic parents briefly mentioned that their family members believed “misinformation” about the COVID-19 vaccine and vaccination more generally. Their family members believed that the COVID-19 vaccine could be dangerous and potentially harm the Hispanic population as a whole. Some of their family members “don’t get vaccinated” in general (e.g., flu shot). However, multiple family members dying from COVID-19 convinced a few family members that vaccination was necessary. 

Overall, parents were quick to point out when they did not always agree with or share the same beliefs as their friends or family members. Instead, these divergent views of vaccination made parents avoid discussing vaccines or even avoid those people in general. 

### 3.5. Information Environment and Trusted Sources 

Parents accessed COVID-19 information online, on their cell phones, or from television (TV). They recalled seeking information through Google searches, YouTube, and state and federal websites to find updated information on vaccination as well as social distancing and masking guidelines and testing. Local city or state text message systems were another commonly used source of information that parents relied on for new details regarding COVID-19 vaccination appointments. TV commercials, the news and Telemundo (Spanish news channel) were also mentioned. However, English-speaking parents felt that the news presented biased, pro-vaccine information, rather than more neutral facts that could have better assisted parents in decision making. A few felt that printed materials were more reliable because they could be referenced later and assist in settling conflicting information from multiple sources. 

Community spaces such as public libraries and schools provided helpful education materials on COVID-19 vaccination and appointment scheduling. In fact, one parent recounted a workshop with a “large turnout” that she helped to organize at her daughter’s school, during which a local pediatrician presented facts about COVID-19 and vaccination and answered parents’ questions. A few other parents found attending similar workshops at their adolescents’ schools helpful. 

Healthcare authorities and healthcare providers such as pediatricians and nurses were commonly cited as trusted sources of health and vaccine information. Most parents used the CDC and FDA websites, commercials, posters, and pamphlets to inform their vaccination decisions, although trust in these authorities waned during the pandemic. Pediatricians and sometimes parents’ own primary care providers were considered the gold standard for trustworthy information for all vaccines including COVID-19. Obtaining information from doctors who represented the community was also strongly valued:

“If it was available, and I had Black doctors who would answer some questions for me, I would be more likely to have my child vaccinated [...] So Black doctors who I trust and who are from the community, like people who actually—I feel like people who are invested here, who grew up [here]—those medical professionals. Or if I could get direct information from them, like no BS, just straight, and they give me the information that I felt was safe, then I would totally do it.”P09, 9 parent of a 10-year-old in August 2021

### 3.6. Historical, Governmental, and Political Contexts Sow Distrust 

Parents had mixed feelings about the motivation and trustworthiness of certain groups involved with developing, promoting, and distributing the COVID-19 vaccines. Namely, they “don’t trust pharmaceutical companies” because they “benefit financially from vaccines.” However, some parents expressed specific concerns about different manufacturers: 

“I feel very confident in them. I am vaccinated and boosted. And my daughter’s vaccinated and boosted. We chose Pfizer over Moderna and Johnson & Johnson. Just more so because it’s the first pharmaceutical name we know and trust […] I didn’t choose Johnson & Johnson, even though it is another trusted name. I feel the one shot was a lot, as opposed to Moderna and Pfizer being broken into two. I think it was two or three weeks apart. My other hesitation was the reports of blood clots in young women. That worried me. And of course, it worried me for my daughter as well. So, we went with Pfizer.”P13, parent of a 13-year-old female in January 2022

Similarly, politicians’ motivations were questionable because they were pushing people to “get back to work and get the economy back.” We analyzed these perspectives by reported political descriptors and did not observe differences by political ideology, though many identified as neither liberal nor conservative. Despite these doubts, most parents were hopeful that the government prioritized the public’s best interest. For instance, they were skeptical about the rapid development and politicization of the vaccination roll-out: 

“The scientists do not always lead the way. A lot of it is political. And so you have to be vigilant, as parents, just people in the community, in general, you have to be vigilant about what it is that you want to accept for your family.”P04, parent of 11- and 9-year-old males in July 2021

Black and Hispanic parents expressed mistrust in the government and medicine, particularly related to the historical mistreatment and unethical testing on certain groups in the US. Multiple participants mentioned that their families and friends believed that the government was currently capable of using vaccinations and other medications to target racial minority groups and cause harmful side effects such as infertility. The trustworthiness of the messenger was particularly important for vaccination messaging among these parents. The Tuskegee Study [33] and the case of Henrietta Lacks [34] were mentioned as reasons that COVID-19 vaccination messaging must be culturally sensitive to vaccine-hesitant individuals within the Black community:

“I think that the conversation around vaccine hesitancy for Black communities really should come from a place of understanding, […] of meeting people where they were, instead of just talking down to them like: ‘that was a long time ago, we’re all good now.’”P02, parent of an 11-year-old female and 13-year-old male in May 2021

Although all participants expressed some hesitancy towards vaccination, Spanish-speaking parents trusted the government and accepted the vaccine more readily than English speakers. Participants acknowledged that they had learned more about the CDC and FDA and their roles as healthcare authorities as a result of the pandemic. 

### 3.7. Poor COVID-19 Communication Exacerbated Concerns about Vaccination

Parents expressed confusion and frustration toward the CDC guidelines, specifically about why and how policies seemed to change so frequently. Although confusion about masking and social distancing was more frequently mentioned than about vaccines, parents understood the need for multiple prevention strategies. In fact, they felt that authorities such as the CDC should have disseminated more accessible prevention information in lay terms. Concerns about the expedited vaccine development and approval process, combined with the inconsistent and changing recommendations from healthcare authorities, exacerbated parents’ expression of hesitancy: 

“I don’t know how I feel right now about the FDA, honestly. […] COVID has changed the way I think about the CDC [and] hospitals. […] I scheduled my son to get vaccinated about three times, and I missed all three appointments because I was still very unsure whether or not I should get him vaccinated […] I feel like the FDA—I feel like they could’ve done more, and provided us with a bit more information.”P11, parent of a 13-year-old in August 2021

In contrast, some parents, mostly non-US-born, tolerated the changes and updated recommendations as indications of transparency of the research process. These parents endorsed the fact that science requires trial and error to produce effective guidelines and vaccines—“that’s how it works, unfortunately”. Another parent was understanding toward the communication and vaccine roll-out: 

“So, these are the people that we rely on for information. So, not to say that they’re getting it right because there were periods of time that you couldn’t even trust what the FDA was saying either […] I mean, I think that the FDA is run by humans, and when you have humans, you have human error. So, of course—I mean, that is the agency that we have to trust in when it comes to certain things. […] I think they’re reliable, you know, but there is a thin line in that error of margin.”P14, parent of 11- and 13-year-old males in January 2022

### 3.8. Parents Had Strong Beliefs in Vaccine Efficacy 

Despite reservations about safety and communication frustrations, participants still believed that the vaccine effectively reduced COVID-19 cases, symptoms, hospitalizations, and ultimately deaths. They acknowledged the benefits for vaccinated individuals, weighed them against the risks, and emphasized the consequences associated with not vaccinating. 

“But truth be told, the COVID vaccine has really, really helped Americans and the world at large because I’ve gone to [unvaccinated] households and seen they have been more terrible, more disastrous. So, the COVID vaccine actually helps and reduces a drastic effect of COVID itself. So, the COVID vaccine came at the right time, it’s actually really helpful, not 100 percent though, but it’s really, really, really helpful.”P08, parent of a 9-year-old female in August 2021

### 3.9. Perceived Control about Vaccination Decision Making

Participants’ perceived control over COVID-19 vaccination differed, with more hesitant parents feeling that authorities were forcing them to accept the vaccine, whereas more accepting parents expressed that choosing vaccination during the pandemic was the best option to protect their families: 

“And I just feel we are being pressured to get it regardless of not wanting to have it. If you want to travel, you have to have your vaccine. If you want to do this, you have to have a vaccine. So even if we did say, no, we don’t want to have it, we’re going to end up having it anyway, period. It’s been forced to us.”P01, mother of a 10-year-old male in May 2021

We also documented differences in views of school- and government-mandated vaccines. For example, most parents were accepting of school mandates for other vaccines, and a subgroup also wanted the COVID-19 vaccine to be required for school. 

“They should do that with the COVID thing. Mandate everybody to get those vaccines […] I mean I’m trying to make sure my daughter has everything that she need to be protected and to avoid a spread to all the kids. I expect the same thing from the rest of the kids. So they should, they should mandate it. And if not, then they need to go to school that everybody’s not vaccinated.”P06, parent of a 9-year-old female in July 2021

As time went on and COVID-19 vaccination became normalized, a couple of parents’ perceptions of safety were influenced by medical professionals and politicians being vaccinated:

“And that also gives you hope because you’re like, hey, if our doctors are getting vaccinated, and all these—everyone working in the—majority of the people working in the medical field, they’re getting vaccinated, and why—and all these politicians, all our presidents, previous presidents. I mean, why wouldn’t I get vaccinated? Why shouldn’t my son get vaccinated?”P11, mother of a 13-year-old male in August 2021

## 4. Discussion

This study identified how parents’ perceptions of COVID-19 and concerns about vaccine safety influenced hesitancy and vaccination behaviors. Our sample was highly accepting of vaccination, and many intended to or already had vaccinated their children, despite expressing concerns about side effects and the vaccines’ novelty. Some interpersonal relationships were sources of misinformation, though parents trusted their healthcare providers and their children’s pediatricians over family and friends. Parents trusted information from local, state, and federal healthcare authorities, although this trust changed over time and many participants expressed frustrations about poor communication from the CDC in particular. 

Overall, parents were accepting of COVID-19 vaccination for their children. We observed stronger acceptance among parents from our second group of interviews, post EUA expansion and ACIP recommendations for younger children. In other words, as time went on, COVID-19 vaccination increased and parents noticed that side effects were minimal. These factors and the expanded age eligibility influenced parents’ vaccination decisions, as our study and others demonstrate [42]. These findings align with national surveys showing a significant increase in parental vaccine acceptance from November 2021 to January 2022 [43]. Similar to our sample, the majority of parents in February 2022 reported that their adolescent had been vaccinated, and only 23% reported intentions of refusing [43]. 

We identified that parents’ key concerns about COVID-19 vaccines were safety and side effects, novelty, and medical mistrust. Parents’ perceived risk of vaccine-preventable diseases [44] and concerns about the safety, necessity, and efficacy of COVID-19 [45,46] and other adolescent vaccines have been well documented [47,48]. We noted that our non-US-born, Hispanic participants were more accepting of vaccination than Black parents and some even supported vaccine mandates. Other studies among US parents have also demonstrated differences by race, ethnicity, and nationality [49,50,51]. Specifically, Szilagyi and colleagues found that Hispanic parents were less hesitant about the HPV vaccine [48] and that Hispanic parents were associated with a higher likelihood of accepting the COVID-19 vaccine [51] for their child compared to other racial and ethnic groups. 

The expedited COVID-19 vaccine development process raised doubts among our participants about the safety and side effects associated with accepting the vaccine, which other researchers have also noted [51,52,53,54]. However, unease about any vaccine that is perceived to be new, such as COVID-19 and HPV, regardless of development timelines, was also common in our sample. The safety and side effects of a novel vaccine are documented primary concerns among parents nationwide [51]. Our sample expressed more confidence and motivation because of the vaccine’s efficacy, whereas Kreuter and colleagues identified concerns about the vaccine having little recorded performance [55]. However, this difference may be due to data collection timing and trial, FDA, and ACIP updates. Nonetheless, our findings and others from earlier pandemic months support variations in hesitancy over time and illustrate that vaccination attitudes and behavior may change over time. Providing parents with tailored vaccine information addressing their concerns may build trust, foster acceptance, and promote vaccination [56], and healthcare providers are uniquely positioned to address parents’ concerns [57]. 

Parents in our study relied on multiple information sources that they considered trustworthy for COVID-19 vaccination education. In contrast to others’ findings that relatives are trusted information sources [58] and that the opinions of family and friends influence acceptance [25], we found that parents from our study discussed COVID-19 vaccination with friends and family but trusted healthcare providers and healthcare authorities over opinions from their personal network. High-quality provider recommendations to vaccinate are strongly associated with adolescent vaccination in general [59,60,61,62]. Another study in NJ among high school teachers also found that primary care providers, in addition to the CDC and state health department, were trusted COVID-19 vaccination sources [63]. Our participants also valued unbiased, scientific evidence regarding vaccination so that they could personally weigh the benefits and risks [58]. We also noted some influence from personal connections with people who experienced COVID-19 hospitalization and death motivating vaccination [46,64,65]. Taken together, these results support multi-level and community-based approaches for disseminating vaccination information [66], including events, workshops, and printed materials at local community health centers, places of worship (i.e., churches, mosques), and schools. 

We also found that the broader social, governmental, and political context contributed to changing attitudes and distrust throughout vaccine development, distribution, and promotion. Similar to the Black parents in our sample, Black adults have expressed medical mistrust in New Jersey and across the country [52,67]. Medical mistrust has also contributed to parental vaccine hesitancy, especially among racially/ethnically diverse populations [68,69]. There was some criticism about politicians’ and the government’s vaccine promotion and roll-out, which was also documented in other states as contributing to public confusion, vaccine hesitancy, and low vaccination [70]. However, we did not observe patterns by political ideology, but other studies have found strong differences in COVID-19 opinions and vaccination behaviors by political affiliation [58,71,72]. 

As with all studies, this qualitative analysis had limitations. Due to social distancing guidelines and other in-person restrictions because of the pandemic, much of our initial recruitment took place virtually, which posed challenges. We also recruited from one major metropolitan area in NJ, which limits the generalization of these findings geographically. Although any parent/guardian was eligible, the parents enrolled in our sample generally had positive attitudes toward vaccination; however, the COVID States Project: A 50-State COVID-19 Survey [73] revealed that mothers are more reluctant to vaccinate their children compared to fathers. Therefore, it is possible that fathers could have offered different perspectives on COVID-19 vaccination for their adolescents, although we did not have any spousal discrepancies reported by our participants. Though we were able to make some comparisons over the course of the pandemic and vaccination approvals, we did not conduct a longitudinal study and only had single data points from each participant. Despite these limitations, we recruited a diverse sample and captured parents’ views and experiences about COVID-19 vaccination, which can inform future efforts to educate parents about vaccination. 

We used this work to inform a collaboration with multiple local organizations in the Greater Newark Area to design and implement COVID-19 Vaccination Ambassador Training during the summer 2021 Delta variant outbreak [74]. We trained public health students, community leaders, and canvassers as Vaccination Ambassadors to disseminate accurate COVID-19 information and promote vaccination appointments. The goal of the training was to prepare Ambassadors to engage with unvaccinated community members, address hesitancy concerns and barriers to vaccination, and schedule COVID-19 vaccine appointments or promote same-day vaccination at mobile vaccination sites across Newark. Through individual outreach, mobile vaccination days, and campaign events over a two-week period, Ambassadors canvassed 81 businesses, conducted 44 presentations, and engaged with 734 community members. Ambassadors counseled unvaccinated community members and distributed incentives to those who made appointments and received vaccines. Over half (62%) of the community members that Ambassadors spoke to had two doses of their vaccine (considered up to date at that time). Of those who were not fully vaccinated, the most common (52%) reason was needing more information. 

## 5. Conclusions

This analysis identified contributors to COVID-19 vaccine-specific hesitancy and motivation among a racially/ethnically diverse sample of parents with adolescents. Despite some age-related concerns, parents were highly accepting of vaccination for themselves and their adolescents. Trust in national authorities changed over time, but overall parents relied on multiple information sources at the local, state, and federal levels from healthcare authorities and providers. Future vaccination messaging and interventions to increase vaccine confidence should leverage local resources and trusted healthcare providers to provide credible information in clinical settings and community spaces such as community centers, schools, and public libraries. To promote future COVID-19 boosters or novel vaccinations, public health campaigns should consider addressing the historical medical mistreatment of minority groups and involve local providers from the community to promote the safety and effectiveness of vaccination.

## Figures and Tables

**Figure 1 vaccines-11-01096-f001:**
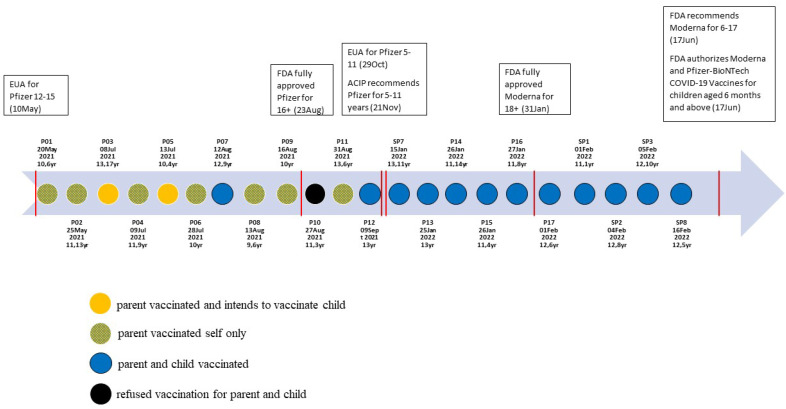
Interview timeline with COVID-19 vaccine authorization and approval dates and participant vaccination behaviors.

**Table 1 vaccines-11-01096-t001:** Sample characteristics of parents of adolescents from Greater Newark Area (*N* = 22).

Parent Characteristics	N (%)
Female	22 (100)
Race/ethnicity	
Non-Hispanic Black	10 (45)
Hispanic	9 (41)
Non-Hispanic Asian	2 (9)
Non-Hispanic White	1 (5)
Education	
Less than high school	4 (18)
High school graduate	1 (5)
College graduate	17 (77)
Language of interview	
English	17 (77)
Spanish	5 (23)
Non-US-born	12 (55)
Political ideology	
Liberal	9 (41)
Moderate	5 (23)
Conservative	1 (5)
Neither	7 (31)
Religion	
Christian	10 (45)
Catholic	5 (23)
Muslim	3 (14)
Agnostic/”NA”/None	4 (18)
# of COVID doses	
0	1 (5)
1	3 (14)
2	15 (67)
3	3 (14)
**Adolescent’s characteristics**	
Age (mean)	11
Male	13 (59)
Insurance	
Private	8 (36)
Medicaid/CHIP	11 (50)
Uninsured	3 (14)
# of COVID-19 doses	
0	10 (45)
1	6 (27)
2	5 (23)
3	1 (5)

**Table 2 vaccines-11-01096-t002:** Exemplary quotes of factors affecting COVID-19 vaccine hesitancy and vaccination among parents.

Risk appraisal	“I understand that it is the right thing to do. It was worse if she got the disease, and that might have a lot of negative impact on her heath.”—P16
Efficacy/effectiveness	“I’m just grateful that we have it [the vaccine]. People have stopped dying, right, or at least at an alarming rate. My friends have stopped dying at least.”—P12
Safety concerns/side effects	“I didn’t choose Johnson & Johnson, even though it is another trusted name. I feel the one shot was a lot, as opposed to Moderna and Pfizer being broken into two. I think it was two or three weeks apart. My other hesitation was the reports of blood clots in young women. That worried me. And of course, it worried me for my daughter as well. So, we went with Pfizer.”—P13
General vaccine confidence	“We think that vaccines are very important, even more so in this pandemic time.” “Entonces nosotros creemos que, bueno, las vacunas son muy importantes, más en este tiempo de pandemia.”—SP3
Motivation	“When they are giving [the vaccines to] 9 years old, oh yes, I will be one of the first people to have my child vaccinated because I want to keep them safe.”—P08
Intention	“We want to protect our children. So, I would rather that COVID didn’t exist, and we didn’t have to get this vaccine, but I’m certainly going to go ahead and do it.”—P17
Acceptance	And so, if my kids’ school said they needed COVID vaccines to go back, then they’re getting COVID vaccines because they gotta go. So, that’s fine.”—P02

## Data Availability

The datasets used and/or analyzed during the current study may be available from the corresponding author on reasonable request.

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
