# Peer review of "Parents’ Intentions, Concerns and Information Needs about COVID-19 Vaccination in New Jersey: A Qualitative Analysis"

_vaccines, 2023, doi:10.3390/vaccines11061096_

Round 1
Reviewer 1 Report
Thank you for the opportunity to review this manuscript - please see attached file with proposed actions to enhance understanding and dissemination of findings.

Author Response
Response to Reviewer 1 Comments
Thank you for these helpful comments and the opportunity to revise our submission (vaccines-2382741) entitled, Parents’ intentions, concerns and information needs about COVID-19 vaccination: A qualitative analysis. Below we respond to each comment individually and note where to find the corresponding changes tracked in the manuscript file.
Reviewer 1, Title. The title is indicative of the content, and, as the study in situated in New Jersey, USA could be more informative by adding New Jersey to the title. Perhaps directly after COVID-19 Vaccination so becoming COVID-19 Vaccination in New Jersey: A qualitative analysis.
Response. We appreciate this suggestion and added “in New Jersey”. It now reads, “Parents’ intentions, concerns and information needs about COVID-19 vaccination in New Jersey: A qualitative analysis.”
Reviewer 1, Abstract. The abstract is presented with the following sections: Background (lines 11-15); Methods (lines 15-18) ; Data collection and analysis (lines 18 – 20); Results (lines 20 – 31); Conclusions (lines 31 – 36). The different weightings allocated to Results and Conclusions reflect the research undertaken and reported.
Reviewer 1, Key words. The current key words are : Covid-19, vaccine hesitancy; vaccination behavior; health communication; adolescent health. The words selected are relevant to the manuscript and are words that may be selected as search terms on this topic area. Consider whether to add parents/guardians.
Response. Thank you for this suggestion. We added “parents” to our list of key words.
Reviewer 1, Introduction. The introduction outlined vaccine hesitancy generally and then more specifically to COVID-19 vaccination. The Introduction identified that the researchers were intending to explore “vaccination concerns and information needs specific to the parents living in the Greater Newark area of NJ, a densely-populated area with a high percentage of historically marginalized groups, including Blacks, Hispanics and other individuals of color, that had low COVID vaccination uptake”.
To add to the reader’s understanding of the importance of vaccine hesitancy, starting the Introduction with a brief outlining what constitutes the significance or “so what” factor would assist. For example, why does vaccination matter for adolescents or perhaps how many adolescent lives do vaccines save for vaccine preventable diseases or even select a disease that was common and caused notable morbidity or mortality which is now avoided by vaccination. This then can add weight to the discussion of social disadvantage addressed later in the manuscript.
Response. Thank you for these suggestions. We added details to the introduction section 1.0, including an explanation of why vaccination matters for adolescents on lines 41-44:
"Vaccines are among the world’s greatest public health advancements and are crucial to the prevention and control of infectious disease outbreaks (1, 2). Childhood vaccination prevented four million deaths in 2021 and another 51 million deaths are projected to be averted through vaccination between 2021-2030 (3, 4).”
Reviewer 1, Methods (including Ethical considerations). This qualitative study was identified as comprising a part of a larger mixed methods study assessing vaccine hesitancy among parents of adolescents in Essex and Hudson counties of Greater Newark.
A qualitative methodology can be an appropriate methodology to elicit perceptions and rationales with participants. In my experience however, it can be beneficial to enhance dissemination of qualitative studies research by adding a little more detail such as the data saturation [See for example Saunders, B., Sim, J., Kingstone, T., Baker, S., Waterfield, J., Bartlam, B., Burroughs, H., & Jinks, C. (2018). Saturation in qualitative research: exploring its conceptualization and operationalization. Quality & quantity, 52(4), 1893–1907. https://doi.org/10.1007/s11135- 017-0574-8 ], exact interview prompts rather than a general description of the topics [“... semi-structured interview guide included a COVID-19 section comprised of questions related to...vaccination attitudes, beliefs, and behaviors or intentions of participants for themselves and their families], and the benefits of using researchers familiar with interviewing, and with the capacity to do so in English or Spanish based on each participant’s preference. This may encourage researchers, students and health professionals more familiar with quantitative methods to read your manuscript and to consider/use information from it. Further, it may encourage researchers with skills in mixed methods and qualitative methodologies to consider using your methods in their location and/or with others with different experiences of social disadvantage and health literacy. I encourage the authors to consider their widest audiences since one goal of research to my perception, is to inform, inspire and expand the body of applicable evidence, widely for impact.
Response. We appreciate this suggestion and added details on lines 135-139, methods section 2.3 Data collection and analysis: “Parents were invited to a HIPAA-compliant Zoom room where female study staff trained in anthropology and health psychology (KC and RW) conducted individual interviews in English or Spanish based on each participant’s preference, which facilitated a diverse variety of parent perspectives. Data collection ended when data saturation was reached, when no new emerging themes occurred during interviews (38).”
We also revised lines 142-146 in methods section 2.3, data collection, to include examples of interview prompts “Examples of the questions asked include: 1) Where/who do you go to for information about COVID-19 vaccination?, 2) Has your adolescent received the COVID-19 vaccine? Why or why not? Do you plan to vaccinate them?, 3) How have your views of vaccines changed since the beginning of the COVID-19 pandemic?, 4) What do you think about the government’s role in COVID vaccination?”
Reviewer 1, Results and Analysis. The results section Table 1 first characterised the participants by gender, race/ethnicity, place of birth, educational attainment, children living at home, and political ideology. Table 2 identified disclosures of COVID-19 vaccination behaviours, intentions and the relevant contextual factors, supported by exemplar quotations.
Participants views and information were then presented about concerns, information and misinformation, the environment and those sources of information which were identified by participants as credible and trusted plus those sources that were viewed with concern and/or mistrust.
Response. Thank you for your feedback. We corrected Figure 1., which had an extra black circle in the middle of the image, by replacing it with the same JPG file used in our original manuscript submission. We ask the publishing/editing staff to confirm the image quality is sufficient.
Reviewer 1, Discussion. The discussion section is an important one as it reviews and contextualises the findings of the research within the field of research. To my perception, of arguably greater importance is the role of identifying the “so what” factor i.e., the significance or implications of the findings.
The manuscript Discussion section identified that participants trusted information from local, state, and federal health care authorities, and, noted that those perceptions differed across time, and that some communications were perceived to be more valuable than others. The changing perceptions were disclosed as being more evident in the second batch of interviews when more information about COVID-19 safety became available, and the context had altered.
The study identified that parents’ key concerns about COVID-19 vaccines were safety, risk, side effects, the perceived uniqueness of COVID-19 as a virus, and mistrust of some sections of the medical bodies such as pharmaceutical manufacturers. The issues identified are consistent with the data elicited, however not all data may have been so relevant such as insurance coverage, political ideology, gender of the adolescent, number of children at home. If these variables were the “screening” variables, then it could be helpful in Table 1 to identify these as such with a superscript identifier.
Response. We did not screen according to insurance coverage, political ideology, or gender of the adolescent; these variables were included for analysis by different sub-groups and provide context about the sample. We removed the variable for number of children in the home or “Mothers with multiple aged children” in Table 1, page 6.
Reviewer 1, Conclusions. The study conclusions identified that the research established contributors to COVID-19 vaccine-specific hesitancy and motivation to vaccinate among a racially/ethnically diverse sample of parents with adolescents in an area within New Jersey. Further, they proposed that any future vaccination messaging and interventions to increase vaccine confidence, based on this study’s results, should incorporate local resources and trusted, credible healthcare providers to disseminate relevant information in clinical settings and community spaces such as community centres, schools, and public libraries. This is an insight that may be especially beneficial in diverse and socially disadvantaged communities.
Reviewer 1, References. Sixty-four references are cited, of which the vast majority are recent publication as might be anticipated with COVID-19. In addition, some additional references may have provided further context to enhance the significance of this study’s findings by comparison more broadly across the United States, by State or globally- for example the following:
Response. Thank you. We added suggested references to introduction lines 57-58 and the other suggested references to discussion lines 448-449, 456-458, and 471-473 to enhance the relevance of our findings.
We also corrected reference number 43:
Anuforo, B., McGee-Avila, J.K., Toler, L. et al. Disparities in HPV vaccine knowledge and adolescent HPV vaccine uptake by parental nativity among diverse multiethnic parents in New Jersey. BMC Public Health 22, 195 (2022). https://doi.org/10.1186/s12889-022-12573-7
We hope that these revisions adequately address the issues the review raised. We believe the updated manuscript is much stronger after making the suggested changes and hope you deem it worthy of publication in Vaccines. We look forward to your response. Thank you.
Reviewer 2 Report
The work carried out with the qualitative research methodology well describes the cognitive needs of the parents interviewed.
In the conclusions it is also good to include the needs of doctors who can play a key role in spreading the correct knowledge, which is the basis of trust in vaccinations.
It is necessary to better describe the method of qualitative research (ethnography, narrative, phenomenological, grounded theory, and case study
It is necessary to restructure table 1 and standardize it: it is not highlighted which is the title of the item whose frequencies are reported and Adolescent characteristics must be in bold and separated by a line a as Parent characteristics
Author Response
Response to Reviewer 2 Comments
Thank you for the opportunity to revise our submission (vaccines-2382741) entitled, Parents’ intentions, concerns and information needs about COVID-19 vaccination: A qualitative analysis. Below we respond to each comment individually and note where to find the corresponding changes tracked in the manuscript file.
Reviewer 2, Conclusions. In the conclusions it is also good to include the needs of doctors who can play a key role in spreading the correct knowledge, which is the basis of trust in vaccinations.
Response. Thank you.
Reviewer 2, Methods. It is necessary to better describe the method of qualitative research (ethnography, narrative, phenomenological, grounded theory, and case study.
Response. We appreciate your feedback. We added more details to lines 148-162 in methods section 2.3, data collection and analysis:
All recordings were professionally transcribed, de-identified and verified for accuracy by research staff. The multidisciplinary study team reviewed transcripts and met regularly to discuss analysis. Two bilingual staff (KC and JV) fluent in Spanish analyzed the Spanish transcripts and then translated key text for combined analysis in English. All transcripts were double-coded by team members (RW, KC, JV) in NVivo 2020 and thematically analyzed using an iterative, immersion-crystallization approach (39, 40). The analytic team (REK, RW, KC, JV) conducted multiple rounds of reading transcripts, writing summaries, identifying themes, coding and organizing text, reflecting on bias and experience, synthesizing patterns, and corroborating interpretations. The analysis was guided by the conceptual frameworks to identify how thoughts and feelings influenced intentions and behaviors (35) and how complex factors across multiple socioecological levels impact vaccine development, promotion, and delivery (41). Themes emerging from the data were also identified and included as new analytic codes/concepts. The full study team with additional experts in nursing, communication, and public health (RB, KG, LK) also discussed themes and patterns, making comparisons and checking assumptions.
Reviewer 2, Table 1. It is necessary to restructure table 1 and standardize it: it is not highlighted which is the title of the item whose frequencies are reported and Adolescent characteristics must be in bold and separated by a line a as Parent characteristics.
Response. Thank you for this suggestion. We restructured table 1 on pages 5-6, results section 3.1. to highlight variable titles by aligning them to the left and indenting variable labels. We also bolded Adolescent characteristics, and separated Adolescent characteristics with a line to match Parent Characteristics.
We hope that these revisions adequately address the issues the review raised. We believe the updated manuscript is much stronger after making the suggested changes and hope you deem it worthy of publication in Vaccines. We look forward to your response. Thank you.